# Debris Flow Infrasound Recognition Method Based on Improved LeNet-5 Network

**Xiaopeng Leng [1], Liangyu Feng [1,*], Ou Ou [1], Xuelei Du [1], Dunlong Liu [2] and Xin Tang [3]**

1   College of Computer Science and Cyber Security (Oxford Brookes College), Chengdu University of Technology, Chengdu 610059, China
2   College of Software Engineering, Chengdu University of Information and Technology, Chengdu 610225, China
3   Chenglizhiyuan Technology (Chengdu Co., Ltd.), Chengdu 610059, China
*   Correspondence: thisisfly0225@outlook.com; Tel.: +86-159-2812-8308

**Abstract:** To distinguish debris flow infrasound from other infrasound sources, previous works have used one-dimensional infrasound shapes and parameters. In this study, we converted infrasound signals into two-dimensional signal time–frequency graphs and created a time–frequency graph dataset containing five common kinds of infrasound. We used deep learning to distinguish debris flow infrasound from other infrasound and improved the deep learning model to enhance the accuracy of debris flow infrasound identification. By improving the LeNet-5 network, we obtained an infrasound signal recognition method for debris flows based on deep learning. After signal preprocessing and model training, this method was able to differentiate target infrasound from environmental infrasound, and a debris flow infrasound recognition accuracy of 84.1% was achieved. The method described in this paper can effectively recognize debris flow infrasound and distinguish it from other environmental infrasound. By such means, more accurate and more timely debris flow disaster warnings may be obtained.

**Keywords:** debris flow; infrasound; neural network; image recognition; deep learning

## 1. Introduction

Geological disasters, especially debris flows, are becoming increasingly serious. In China, more than 810 debris flow disasters occur every year [1]. Geological disasters, including debris flows, cause nearly 2100 casualties and economic losses of nearly USD 600 million every year. Debris flows have also seriously affected sustainable development. The harmful impacts of debris flows are now of concern to countries across the world.

Infrasound refers to sound waves with a frequency lower than 20 Hz, with characteristics of low frequency, long wavelength, easy diffraction, and small attenuation by the viscous effects of atmosphere and water. Infrasound can be transmitted over long distances in the air. Infrasound waves are also excited when debris flows occur, and their propagation speed in air at room temperature is dozens of times that of debris flows themselves. Therefore, by identifying the infrasound signals excited by debris flows, information on debris flow events can be obtained in advance, and early warning procedures activated. In recent years, the infrasound monitoring and early warning of debris flows have been widely recognized as priority areas for development. Zhang Shucheng et al. [2] developed a debris flow infrasound alarm system using core sensors, which send an alarm 10 to 30 min before the onset of a debris flow event. Chou et al. [3] observed and analyzed a debris flow in Jiangjiagou. They found that its main frequency range was 5~15 Hz, and concluded that the interference of environmental noise caused by phenomena such as wind and rain should be taken into account for practical application purposes. Hüb et al. [4] also conducted a debris flow infrasound observation experiment in Jiangjiagou. They reported a peak value of debris flow infrasound between 8 Hz and 12 Hz, with sound pressure levels typically lower than 4 Pa. Liu Dunlong et al. [5] proposed a new method for identifying

debris flow infrasound signals from the spectral, duration, and waveform characteristics of the signal. Li Chaoan et al. [6] developed an infrasound monitor to monitor debris flow ditches along a railway.

In light of this previous research, it can be said that the main frequency band of debris flow infrasound waves is relatively wide, and easily mixes with many environmental interference noises. Filtering and similar types of processing by single-chip microcomputers equipped for monitoring purposes cannot entirely eliminate the interference of environmental noises. Previous studies have used filtering, amplification, statistical feature calculation, correlation analysis, and similar processing techniques on time-domain signals. However, the infrasound signal of a debris flow is random. One-dimensional characteristic values such as mean and variance can only reflect the statistical law of the signal amplitude, and cannot automatically extract features for learning from various infrasound signals. For these reasons, they are of limited value in practical applications due to the variations among different debris flow environments [7].

The propagation speed of infrasound waves in air at 25 °C is about 344 m/s. This is more than ten times the movement speed of debris flows (5~20 m/s). By identifying the infrasound triggered by a debris flow, the onset time of a debris flow event can be estimated in advance, and valuable warning time can be obtained.

In addition to debris flows, other physical phenomena such as earthquakes, sliding waves, and lightning may also trigger infrasound. Traditional time–frequency analysis does not have the ability to identify and distinguish between signals with different time–frequency characteristics.

In light of the above, this paper describes a method different from those used in previous studies which sought to distinguish debris flow infrasound from other infrasound using one-dimensional infrasound shapes and parameters. Using the method reported here, we converted infrasound signals into two-dimensional signal time–frequency images and created a time–frequency image dataset containing five common infrasound patterns. Considering the strong ability of deep neural networks to extract image features, we designed a method which incorporated deep learning for the recognition of debris flow infrasound signals based on an improved LeNet-5 network. The image recognition method was used to recognize the infrasound signal spectrum. The learning ability of the neural network yielded a better recognition effect, and so, the method described here represents a new and promising contribution to reducing and preventing harm caused by debris flow disasters.

## 2. Methodology

### 2.1. Debris Flow Infrasound Signal Characteristics

During the rapid movement of a debris flow, collisions between particles, between particles and fluids, and between fluids and groove beds or bank slopes all generate infrasonic signals through the medium of air [8]. These signals propagate at a speed of about 344 m/s over long distances with no or little attenuation, enabling alerts in advance of disasters caused by debris flows. By identifying and monitoring debris flow infrasound signals, early warnings can be issued. However, many other natural and anthropogenic phenomena also generate infrasound signals. Sea storms, volcanic eruptions, earthquakes, thunder, and lightning may all be accompanied by infrasound waves. Even some loudspeakers are capable of generating infrasound. For this reason, it is not enough to simply determine whether an audio signal contains infrasound components when carrying out infrasound monitoring for early warning of debris flow events. In debris flow infrasound monitoring, it is crucial to classify and identify the environmental infrasound, so as to accurately distinguish the infrasound signal of debris flow excitation, and thereby obtain early warning of a debris flow event.

To address this problem, we considered the results of previous research on debris flow infrasound, and compared the interference infrasound characteristics of debris flow and some common natural phenomena [9] (Table 1).

**Table 1.** Infrasound characteristics of debris flow and other common natural phenomena.

| Type | Center Frequency/Hz | Energy Concentration Range/Hz | Duration |
|------|---------------------|-------------------------------|----------|
| Debris flow | 10~15 | 5~20 | Very long, at least 16 min |
| Thunderbolt | 5~7 | 0.6~12.3 | Shorter, usually less than 30 s |
| Explosion | 0.5~12 | 0.01~16.7 | Shorter, usually less than 10 s |
| Engine | 6~11 | 0.1~16.8 | Slightly longer, usually more than 30 s |
| Wind | 3~5 | 0.01~9.1 | Long, persistent |

Having determined the specific characteristics of debris flow infrasound signals such as wide center frequency, an energy concentration range of 5~20 Hz, long duration, etc., we designed a method for identifying debris flow infrasound signals based on signal preprocessing and LeNet-5, which uses signal processing methods such as low-pass filtering and wavelet soft-threshold denoising to retain infrasound signals and eliminate environmental noise interference. We then used the improved convolutional neural network LeNet-5 for the following purposes: (1) to automatically extract debris flow infrasound features from the generated infrasound time–frequency image; (2) to accurately identify debris flow infrasound waves; and (3) to provide a reliable basis for debris flow disaster monitoring and early warning.

*2.2. Debris Flow Infrasound Identification Method Based on LeNet-5*

2.2.1. Signal Preprocessing

The instrument used in this study was an infrasound acquisition device developed by the Institute of Acoustics, Chinese Academy of Sciences (Figure 1).

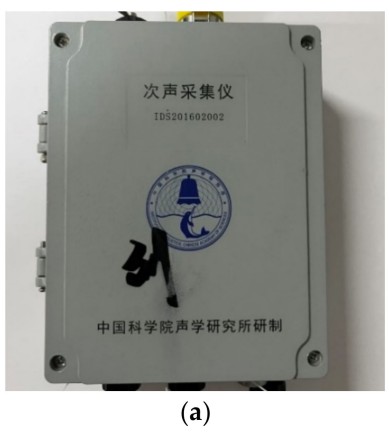 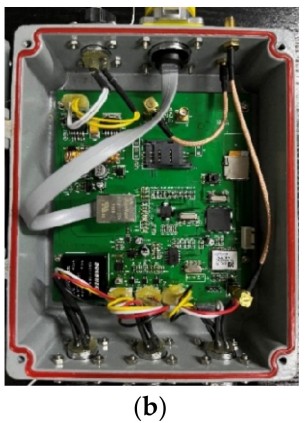 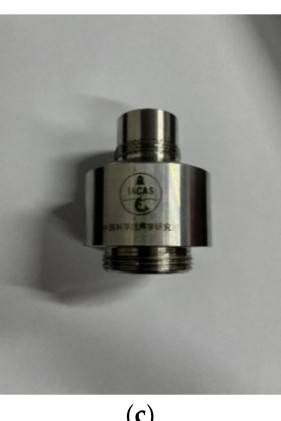

(**a**)        (**b**)        (**c**)

**Figure 1.** (**a**) Infrasound collector developed by Acoustics Research Institute of Chinese Academy of Sciences is written on the external of the instrument; (**b**) instrument internal; (**c**) infrasound sensor.

This infrasound sensor is a new product (IDS2016) of the Institute of Acoustics, with a frequency response range of 2~200 Hz and a sensitivity of 50 mV/Pa. For this study, we set the instrument to a sampling frequency of 100 Hz. Our acoustic wave data came from two sources: (1) the infrasound signals generated by debris flows in the Jiangjiagou watershed of Dongchuan, Yunnan; and (2) infrasound signals generated by collected debris flow in an indoor water tank.

Because the original signal collected by the instrument contained environmental noise in addition to debris flow infrasound, it was necessary to perform preprocessing operations such as filtering and denoising of the collected signal by the following means:

1. Low-pass filtering

To analyze the collected debris flow signal effectively, and to study the characteristics of the signal in each frequency band in sections, it was necessary to design a filter for the signal. Digital filters can be divided into FIR digital filters and IIR digital filters [10].

Compared with the IIR filter, the FIR filter is more stable, and fast Fourier transform can be used for the filtering function, resulting in greatly improved operational efficiency. For this reason, we used the window function method in the FIR filter to design our low-pass filter.

The FIR digital filters designed with different window functions have different filtering effects on the signal and different spectral characteristics. The reference data of common window functions [11] are shown in Table 2.

**Table 2.** Reference data of common window functions.

| Window Function | Sidelobe Peak (dB) | Main Lobe Width | Minimum Stopband Attenuation (dB) |
|---|---|---|---|
| Rectangular | −13 | $4\,\pi/N$ | −21 |
| Triangular | −25 | $8\,\pi/N$ | −25 |
| Hanning | −31 | $8\,\pi/N$ | −44 |
| Hamming | −42 | $8\,\pi/N$ | −53 |
| Blackman | −58 | $12\,\pi/N$ | −74 |

Ideally, the narrower the main lobe width of the window function, the more accurate its frequency resolution, and the closer it is to the ideal spectrum response curve; the smaller the sidelobe, the greater the attenuation, the stronger the ability to filter out interference signals, and the higher the amplitude accuracy. Generally, the reduction in main lobe width comes at the expense of minimal attenuation of the sidelobe [11]. Therefore, for our actual design, it was necessary to select an appropriate window function according to the characteristics of the signal and our actual study needs.

We found strong environmental interference in the initial infrasound signal obtained. We therefore increased the attenuation capacity to analyze the required debris flow infrasound signal; at the same time, we sought to maintain the accuracy of the frequency resolution. After carrying out a comparative analysis, we selected the Hanning window function with larger sidelobe attenuation and a narrower main lobe width to design the FIR low-pass filter.

A characteristic table of the Hanning window function is shown in Table 3.

$$\Delta W = \frac{(W_s - W_p)\,\pi}{44} \tag{1}$$

$$N = \frac{6.2\pi}{\Delta W} + 1 \tag{2}$$

**Table 3.** Characteristic table of Hanning window function.

| Window Function | Precise Transition Bandwidth | Stopband Minimum Attenuation/dB |
|---|---|---|
| Hanning | 6.2 $\pi$/N | −44 |

Equations (1) and (2) are formulae for calculating the filter order, in which $N$ represents the filter order, $W_s$ denotes the stopband cutoff frequency, and $W_p$ is the passband cutoff frequency [11].

To assess our actual design, we set the sampling frequency of the filter $F_s$ to 100 Hz in line with the acquisition frequency of the instrument. Because the frequencies of infrasound signals are less than 20 Hz, we set the passband frequency $W_p$ to 0 Hz and the stopband frequency $W_s$ to 20 Hz, to maintain a frequency below 20 Hz. Using the data in Table 3, and working through Equations (1) and (2), we obtained a calculated filter order figure of 15.

The results of the frequency domain diagram before and after signal filtering are shown in Figure 2.

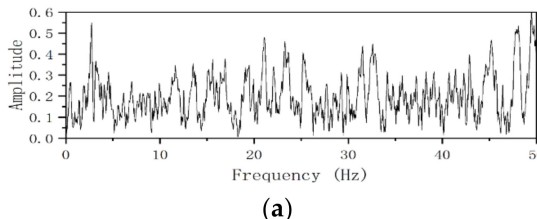 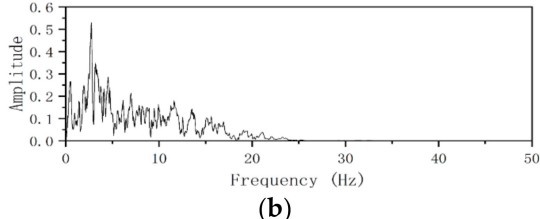

(**a**)        (**b**)

**Figure 2.** (**a**) Original signal spectrogram; (**b**) low-pass filtered signal spectrogram.

In testing, our designed filter exhibited good performance. Signals with frequencies below 20 Hz passed normally, while high-frequency signals exceeding the set threshold were weakened or entirely blocked.

2.       Wavelet Soft-Threshold Denoising

Infrasound signals are nonlinear and non-stationary. Many traditional denoising methods are available, such as Wiener filtering [12] and median filtering, but most of these rely on linear systems, have a limited range of denoising, and have an effectiveness which depends on appropriate state function. However, wavelet transform can perform multi-scale analysis on the signal, separate different frequency bands, and determine a threshold for denoising and reconstruction, which can achieve the separation of signal and noise.

Wavelet transform is widely used in the field of signal processing. Different wavelet bases have different time–frequency characteristics, and different results are obtained when analyzing the same signal with different wavelet bases. In theory, when selecting and constructing a wavelet function, the properties of orthogonality, tight support, symmetry, and vanish-moment order should all be optimized. Orthogonality affects the frequency resolution, tight support ensures positive spatial local properties, symmetry ensures that the wavelet-filtering characteristics have a linear phase shift without causing signal distortion, and the vanish-moment order reflects the concentration of energy. However, these four properties cannot be simultaneously optimized in practice. A larger vanish-moment order results in better wavelet smoothness and increased tight support. The local characteristics of the wavelet analysis cannot be guaranteed, and this is not conducive to the realization of the algorithm. Therefore, each index must be comprehensively considered in the course of the experiment, to assess its suitability for the signal to be processed. The main characteristics [13] of several commonly used wavelet bases are shown in Table 4.

**Table 4.** The main features of several commonly used wavelet bases.

| Function | Orthogonality | Tight Support | Support Length | Symmetry | Vanish-Moment Order |
|---|---|---|---|---|---|
| Biorthogonal | No | Yes | Reconstruct: $2N_r + 1$ <br> Decompose: $2N_d + 1$ | Asymmetrical | $N_r + 1$ |
| Daubechies | Yes | Yes | $2N - 1$ | Approximate | $N$ |
| Symlet | Yes | Yes | $2N - 1$ | Approximate | $N$ |
| Morlet | Yes | No | Limited length | Symmetrical | — |

In this study, when we processed infrasonic signals, we sought to ensure the local characteristics of the signal and to reduce distortion. We found the sym and db wavelet bases to be most in line with our requirements. Both are compactly supported and symmetrical; however, the sym wavelet offers better symmetry and stronger localization ability in the time and frequency domain, and can provide more practical and more specific digital filters of finite length in the process of wavelet decomposition of signals [13]. For these reasons, and after a thorough review of the characteristics of infrasound signals and the principles of wavelet denoising, we chose to use the sym wavelet, which is similar to the infrasound signal waveform, for threshold denoising purposes.

Wavelet threshold denoising can be divided into hard and soft threshold functions [14]. The threshold functions of the two methods are manifested as follows:

Hard threshold function:

$$w'_{(j,k)} = \begin{cases} w_{(j,k)} & \left|w_{(j,k)}\right| \geq s \\ 0 & \left|w_{(j,k)}\right| < s \end{cases} \tag{3}$$

Soft threshold function:

$$w'_{(j,k)} = \begin{cases} sgn(w_{(j,k)})\left(\left|w_{(j,k)}\right| - s\right) & \left|w_{(j,k)}\right| \geq s \\ 0 & \left|w_{(j,k)}\right| < s \end{cases} \tag{4}$$

where $s$ represents the threshold function, calculated thus:

$$s = \sigma \times \sqrt{2 \ln N} \tag{5}$$

where $w_{(i,k)}$ represents the wavelet coefficient, $w_{(j,k)}$ is the processed wavelet coefficient, $\sigma$ denotes the noise standard deviation, and $N$ is the signal length.

In wavelet threshold processing, relevant rules must be selected. Options available include fixed threshold, unbiased risk estimate threshold, heuristic threshold, and extreme value threshold rules. For this study, we adopted the unbiased risk estimate threshold rule. After wavelet decomposition, we performed soft-threshold processing on the coefficients at all levels and finally reconstructed the processed wavelet coefficients.

Compared with the original signal, the SNR (signal-to-noise ratio) value of the signal can reflect the denoising performance to a certain extent. The larger the SNR, the better the denoising effect, which is defined as follows [15]:

$$SNR = 10 \log_{10}\left(\sum_{n=1}^{N} I^2 \Big/ \sum_{n=1}^{N} [I - I_n]^2\right) \tag{6}$$

where $n$ represents the number of sampling points, $I$ is the original signal, and $I_n$ denotes the denoised signal.

A section of the debris flow infrasound signal is taken as an example. The SNR values of different layers were calculated, and the results are shown in Table 5. When the number of decomposition layers was 3, the SNR value was highest, and the denoising effect was best. Considering the characteristics of debris flow infrasonic signals, and the value of the signal-to-noise ratio, we chose to use a sym3 wavelet to decompose the signal into three layers for wavelet soft-threshold denoising.

**Table 5.** Debris flow signal SNR values with different decomposition layers.

| Layers | 1 | 2 | 3 | 4 | 5 |
|--------|------|-------|-------|-------|------|
| SNR | 7.78 | 10.98 | 14.83 | 12.05 | 9.76 |

The effect of wavelet soft-threshold denoising is manifested in Figure 3.

According to Equation (5), the SNR before denoising was 9.90, and the SNR after denoising was 14.83. The interference noise in the debris flow infrasound signal was effectively suppressed.

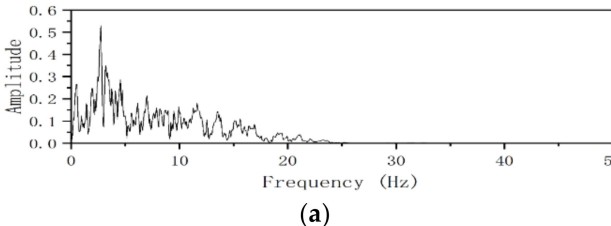 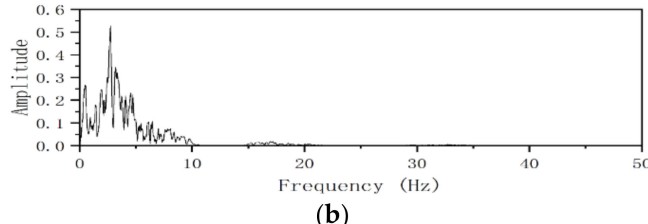

(**a**)                                                       (**b**)

**Figure 3.** (**a**) Filtered signal spectrogram; (**b**) signal spectrogram after filtering and denoising.

### 2.2.2. Infrasound Time–Frequency Image Construction

The time–frequency feature is the key to the identification of debris flow signals. It can expand the infrasound signal from a one-dimensional state to a two-dimensional state and can reveal changes in the infrasound signal more conveniently and intuitively in the time and frequency domains. The commonly used time–frequency representations are divided into two types: linear and nonlinear. Typical linear time–frequency representations include short-time Fourier transform and continuous wavelet transform, and typical nonlinear time–frequency representations include Wigner–Ville distribution [16]. Drawing on the results of previous research, we compared the advantages and disadvantages of these three time–frequency analysis methods, as shown in Table 6.

**Table 6.** Advantages and disadvantages of time–frequency analysis methods.

| Method | Advantage | Disadvantage |
|---|---|---|
| Short-time Fourier transform | Easy to implement, no cross-term interference | Time–frequency focus is limited and cannot adapt |
| Continuous wavelet transform | Strong adaptive ability | Choice of a suitable wavelet basis is arduous |
| Wigner–Ville distribution | High time–frequency focusing characteristics | Strong cross-term interference |

Taking a preprocessed debris flow infrasound signal as an example, a comparison of the effects of the three time–frequency analysis methods is shown in Figure 4. The time–frequency image generated by the short-time Fourier transform method has poor time–frequency focus; for this reason, it is difficult to distinguish the time when the infrasound signal of the debris flow is emitted. The time–frequency image generated by the WVD analysis method has oscillating cross-interference terms, which causes the time–frequency characteristics of the signal to be ambiguous. The time–frequency image generated by the continuous wavelet transform method has clear features and no cross-term interference. In the whole period of 0~15 s, different strengths of debris flow infrasound signal energy can be observed. Between 2s and 4 s, the infrasound signal energy is stronger, with a focus in the frequency range of 5~20 Hz, indicating the occurrence of debris flow at this moment. In light of these findings, we chose the continuous wavelet transform method to construct our infrasound time–frequency image.

### 2.2.3. Recognition Method Based on Improved LeNet-5

The classic convolutional neural network models include LeNet-5, AlexNet, GoogleNet, VGG, and Deep Residual Learning. The total number of layers of these five types of networks [17] are shown in Table 7.

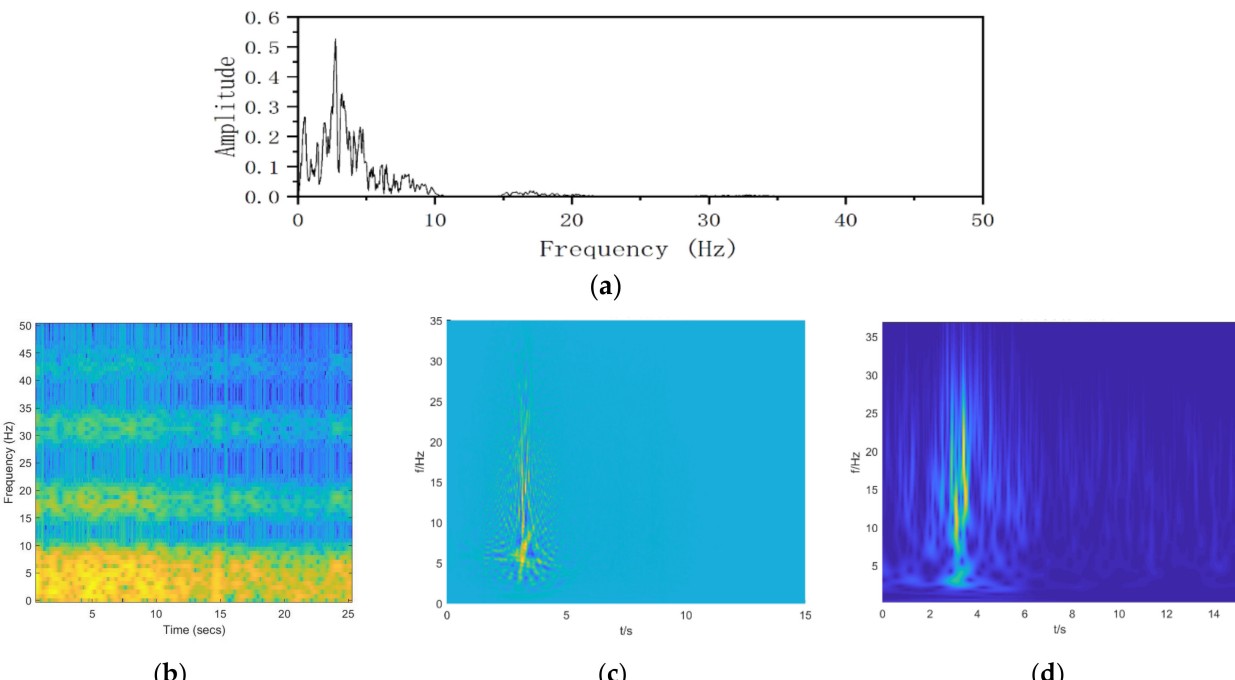

**Figure 4.** (**a**) Spectrogram of infrasound signals of pretreated debris flows; (**b**) short-time Fourier transform; (**c**) continuous wavelet transform; (**d**) Wigner–Ville distribution.

**Table 7.** The total number of layers in five types of classic convolutional neural network models.

| Model | LeNet | AlexNet | GoogleNet | VGG | Deep Residual Learning |
|-------|-------|---------|-----------|-----|------------------------|
| Total layers | 4 | 8 | 11–19 | 22 | 152 |

Because the time–frequency diagrams of infrasound signals studied in this paper are all based on lines and curves, a simple convolutional network can identify the lines, edges, and corners of graphics. For this reason, we adopted the LeNet-5 network with fewer layers and parameters for the purposes of our study.

The LeNet-5 model is a classic CNN model first proposed by LeCun Y. et al. [18]. In its early years of practical application, many banks in the United States adopted the LeNet-5 model to recognize handwritten numbers on checks, and its accuracy was very good. The LeNet-5 model is a self-learning process which involves ensemble learning, back-propagation, and selection optimization, and also combines feature extraction and image recognition. The classic LeNet-5 convolutional network consists of one input layer, two convolutional layers, two pooling layers, two fully connected layers, and an output layer.

To improve the performance of the entire LeNet-5 network for the purpose of infra-sound recognition of debris flow, and to achieve higher levels of accuracy, we made the following optimization improvements:

(1)   We changed the size of some convolution kernels in the network from $5 \times 5$ to $3 \times 3$ to reduce the number of parameters and computational complexity.

(2)   We used ReLU as the activation function.

The classic LeNet-5 model uses sigmoid as the activation function (Figure 5a) and achieves high computational complexity and saturation. When the input is very large or very small, the slope of the image tends to 0, that is, the derivative gradually approaches 0. In such a circumstance, the problem of gradient disappearance is prone to occur. To the underlying network, the gradient becomes very small, thus making it difficult for the network to be effectively trained [19]. However, in the same circumstances, ReLU retains low computational complexity and fast convergence speed. As can be seen from Figure 5b,

when $x \geq 0$, the derivative is 1. Therefore, the ReLU function prevents the gradient from decaying when $x \geq 0$, and this effectively alleviates the problem of gradient disappearance.

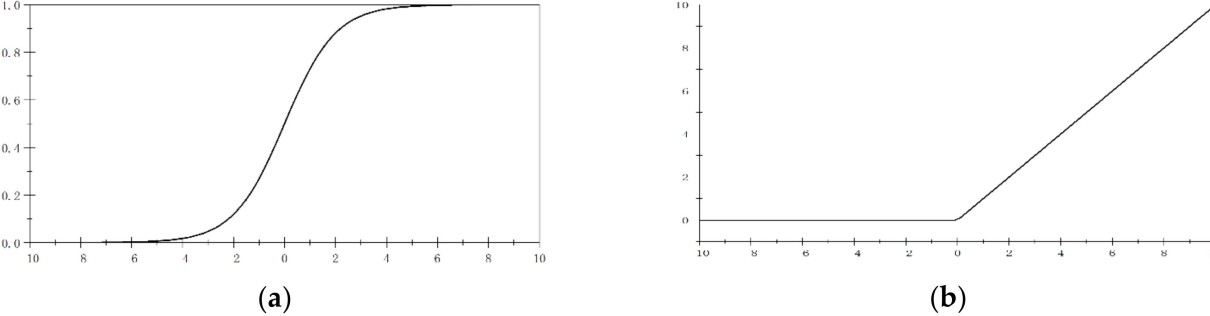

(**a**)    (**b**)

**Figure 5.** (**a**) Sigmoid activation function; (**b**) ReLU activation function.

(3)    We added the Inception_v1 module [20] to the model to increase the network depth while extracting more features of the target.

(4)    We adopted a dropout strategy [21] in the fully connected layer to discard the connection of some neurons with a certain probability, reduce the number of network training parameters, and improve the network training effect.

(5)    In the last layer, we used SoftMax [22] regression as the output layer; the results were mapped to the (0, 1) interval, and the output results were converted into probability problems.

The improved LeNet-5 network structure proposed in this paper is shown in Figure 6. The structure includes an input layer, 2 convolution layers, 3 pooling layers, an Inception module, a fully connected, and an output layer. The network input is an image of $100 \times 100$ pixels. The features of the image are extracted through the convolution layer and the Inception module, and the output layer is classified by the SoftMax classifier. Table 8 describes the specific parameters of the improved LeNet-5 network structure, including the type of each layer, the size of the convolution kernel, the stride, and the size of the output of each layer.

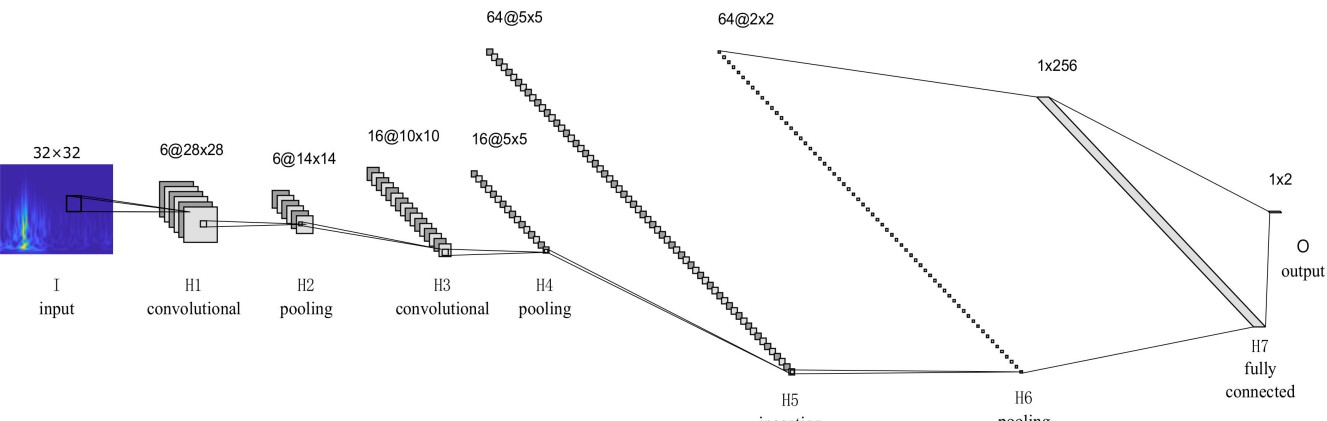

**Figure 6.** Improved LeNet-5 neural network structure diagram.

**Table 8.** Parameters of improved LeNet-5 network structure.

| Layer | Type | Convolution Kernel Size | Step Size | Output Size |
|---|---|---|---|---|
| I | Input layer | — | — | $32 \times 32 \times 1$ |
| H1 | Convolutional layer | $5 \times 5$ | 1 | $28 \times 28 \times 6$ |
| H2 | Max pooling layer | $3 \times 3$ | 2 | $14 \times 14 \times 6$ |
| H3 | Convolutional layer | $5 \times 5$ | 1 | $10 \times 10 \times 16$ |
| H4 | Max pooling layer | $3 \times 3$ | 2 | $5 \times 5 \times 16$ |
| H5 | Inception | $1 \times 1$ | | — |
| | | $1 \times 1; 3 \times 3$ | | — |
| | | $1 \times 1; 5 \times 5$ | | — |
| | | $3 \times 3; 1 \times 1$ | | $5 \times 5 \times 64$ |
| H6 | Max pooling layer | $3 \times 3$ | 2 | $2 \times 2 \times 64$ |
| H7 | Fully connected layer | — | — | 256 |
| O | Output layer | — | — | 10 |

## 3. Results

### 3.1. Experimental Environment

For the experimental training in this paper, we used a R7-5800H CPU computer with an 8 GB GPU memory. The programming language and framework structure were Python3.10 and Pytorch1.11, respectively. We used the PyCharm code-writing platform to write the program. Using Pytorch and related volumes, the integrated neural network library implemented the improved LeNet-5 model. MATLAB and other tools were used to analyze the data obtained after training.

### 3.2. Construction of Time–Frequency Image Dataset

The images used in this study were all time–frequency images converted from infrasound signals collected by the instrument from the actual environment or from indoor experiments after preprocessing. To confirm that signal preprocessing improved the accuracy of identifying debris flow infrasound waves, we constructed two datasets. Dataset1 consisted of 405 pictures with a pixel size of $875 \times 656$, including 81 time–frequency images of debris flow, wind, lightning, rain, and lightning infrasound before preprocessing. Dataset2 comprised preprocessed time–frequency images of the same scale showing debris flow, wind, lightning, rain, and thunder infrasound. During the experiment, we randomly divided the dataset into a training set and a test set with a ratio of 0.8:0.2. Figure 7 shows example images from the two datasets.

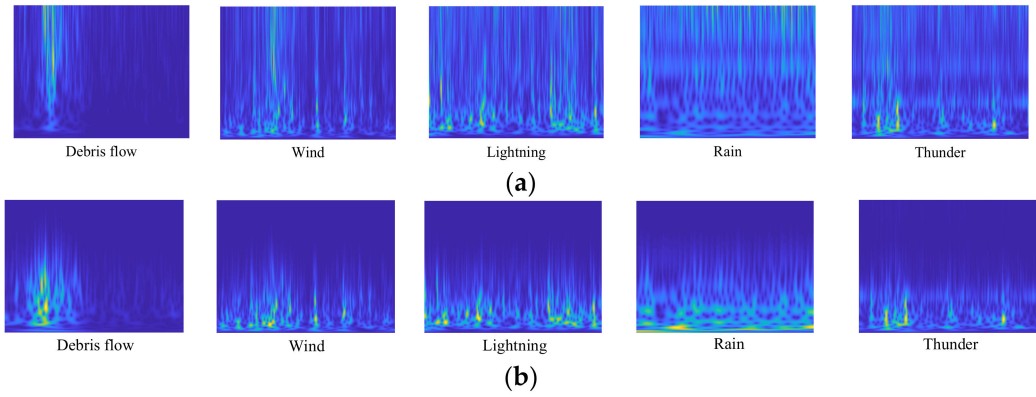

**Figure 7.** (**a**) Partial time–frequency images of Dataset1; (**b**) partial time–frequency images of Dataset2.

### 3.3. Experimental Process and Analysis of Results

To determine the influence of preprocessing on recognition probability, and to verify the effectiveness of our proposed method, we trained our models with the following

three combinations: (1) combination A (proposed method): after preprocessing (dataset2) + improved LeNet-5; (2) combination B: before processing (dataset1) + improved LeNet-5; (3) combination C: before preprocessing (dataset1) + LeNet-5. During programming, different network models adopted the same exponential decay learning rate, objective optimization function, and parameter update method. The accuracy rate was used to evaluate the infrasonic wave identification capability of the three combinations. The loss value was used to further evaluate their merits and demerits. After 1000 iterations of each combination, we obtained comparison charts for recognition accuracy and loss value, and these are shown in Figure 8.

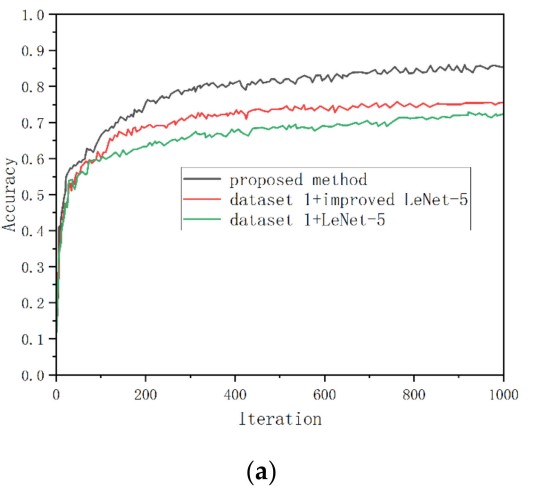

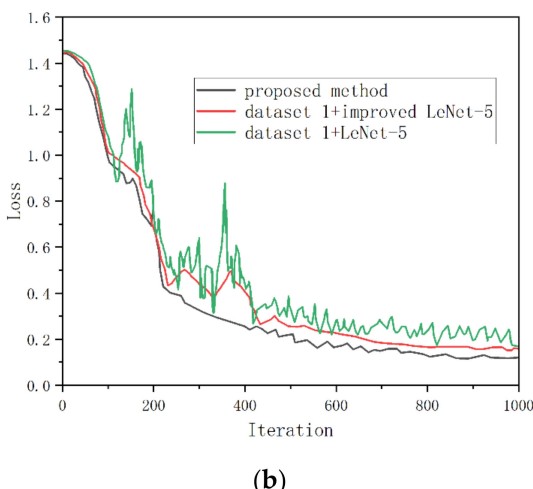

(**a**) (**b**)

**Figure 8.** (**a**) Identification accuracy comparison chart; (**b**) comparison chart of loss values.

## 4. Discussion

It can be seen from Figure 8a and Table 9 that in this experiment, which sought to identify debris flow infrasound signals based on LeNet-5, the classification effect of combination C was lowest, with an accuracy rate of 70.4%. The classification accuracy of combination B was 4.8% higher than that of combination C, at 75.2%, and the classification accuracy of combination A (our proposed method) was 13.7% higher than that of combination C, at 84.1%.

**Table 9.** Statistics of recognition accuracy for each method combination.

| Method | Proposed Method (A) | Combination B | Combination C |
|---|---|---|---|
| Recognition accuracy | 84.1% | 75.2% | 70.4% |

Similarly, it can be seen from Figure 8b that the combination C model exhibited a slow convergence speed and obvious oscillation in the convergence curve, while combination A (the proposed method) exhibited a fast convergence speed and a smooth convergence curve without obvious oscillation. Combination B produced better results in terms of oscillation compared with combination C, but combination A (the proposed method) was more smoothed.

These experimental results demonstrate that the infrasound recognition method of debris flow proposed in this paper has high recognition accuracy and application value, and can effectively serve the purpose of early warning of debris flow events.

## 5. Conclusions

In this study, based on the principle of feature difference between debris flow infrasound signals and environmental interference infrasound, we combined a signal time–frequency map with deep learning, and made full use of the powerful feature extraction

and feature learning of deep learning to design a recognition method for debris flow infrasound signals based on the LeNet-5 network, which represents a new means of preventing and controlling the harm caused by debris flow disasters. Our experimental findings led us to the following conclusions:

(1) This method is innovative, at least to some extent. By using deep learning techniques that are good at extracting features, we overcame the weaknesses of previously used methods of identifying debris flow infrasound based on one-dimensional signal waveforms and parameters.

(2) The recognition accuracy of this method is relatively high. Our experimental results demonstrated the good recognition performance of the system, and thus, confirmed a new and feasible technical approach for the reduction in and prevention of harm caused by debris flow events.

**Author Contributions:** Conceptualization, X.L. and O.O.; data curation, X.D. and D.L.; writing—original draft preparation, L.F.; writing—review and editing, X.L.; supervision, X.T.; project administration, X.L. All authors have read and agreed to the published version of the manuscript.

**Funding:** This research was funded by the Basic Research Project of the Department of Science and Technology of Sichuan Province (No. 2021YJ0335), the National Natural Science Foundation of China (No. 42001100), and the Project of the Department of Science and Technology of Sichuan Province (Nos. 2021YFG0258).

**Institutional Review Board Statement:** Not applicable.

**Informed Consent Statement:** Not applicable.

**Data Availability Statement:** Data can be made available upon request.

**Conflicts of Interest:** The authors declare no conflict of interest.

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
