# Peer review of "Debris Flow Infrasound Recognition Method Based on Improved LeNet-5 Network"

_sustainability, doi:10.3390/su142315925_

Round 1

Reviewer 1 Report

I recommend a full revision of English and phrasing. Unfortunatly the manuscript is hard to read and this could greatly influence the quality of the work.

Introudction is unsatisfactory, the authors just described the result obtained from other auhtors whitout a proper description of the state of art, goal and innovation proposed in this work.

Please, introduce in the scientific background of your study the importance of combining advanced soft-computing and image processing models in the monitoring, management, and forecasting of natural resources over vegetated masses (i.e.,

Wu, J. L., Xiao, H., & Paterson, E. (2018). Physics-informed machine learning approach for augmenting turbulence models: A comprehensive framework. Physical Review Fluids3(7), 074602.

Lama, G.F.C., Errico, A., Pasquino, V., Mirzaei, S., Preti, F., Chirico, G.B. 2022. Velocity uncertainty quantification based on Riparian vegetation indices in open channels colonized by Phragmites australis. J. Ecohydraulics 7(1), 71–76. https://doi.org/10.1080/24705357.2021.1938255.

Please, pay attention to the JOURNAL TEMPLATE in all sections, including tables, references, captions, units, and Figures.

A real discussion miss, I cannot see a real comparison of the result of your work with the other research mentioned in the introduction.

Please organize better the conclusion maybe using a bullet point sistem.

Author Response

Dear reviewer
Thank you for taking the time to read this message. I apologize for my negligence, I found an error in Point3's response. I have corrected it. The corrected reply has been attached and sent to you again. Expresses my apology once more.

Reviewer 2 Report

The manuscript entitled Debris flow infrasound recognition method based on improved LeNet5 network, by X. Leng, L. Feng, O. Ou, X. Du, D. Liu & X. Tang, presents an interesting work.

In general, the manuscript should be acceptable for publication but some serious problems must be repaired prior to publication. It needs some significant improvement. Some suggestions are as follows:

  1. Please follow the journal author instructions. It would be useful for the reader to follow the classical text structure (i.e. Introduction-methodology-results-discussion-conclusions). A better presentation of your results and an extensive discussion would improve your paper.
  2. Please use different terms in the “Title” and the “Keywords”.
  3. The abstract should state briefly the purpose of the research, the principal results and major conclusions. An abstract is often presented separately from the article, so it must be able to stand alone.
  4. I propose to the authors to be more specific, explanatory and simplified in order to be easily understandable from the readers.
  5. It would be useful to be described the aim of this paper.
  6. The English language usage should be checked by a fluent English speaker. It is suggested to the authors to take the assistance of someone with English as mother tongue.
  7. You could enrich the scientific literature. It is so poor.
  8. Please justify convincingly why this manuscript (method, thematology etc) connected with sustainability’s content and scope. Perhaps the using of proper literature would be helpful.
  9. Correct references in the text and the reference list according to the journal’s format. Please format the references’ list by using the correct journal abbreviations. See the following link: https://images.webofknowledge.com/images/help/WOS/A_abrvjt.html
  10. Please be careful with the spaces between the words.

Reviewer 3 Report

In this paper, very useful debris flow infrasound recognition method was proposed and verified. 

The manuscript is very well written, but there are some errors in the paper, so please correct them.

p1 line34 " It has the characteristics of low frequency, long wavelength, easy diffraction, strong penetrating power" The ultrasound waves have a much greater penetrating power than infrasound. Is it true that infrasound has strong penetrating power? 

6 194  There are errors at the threshold value condition in Eq. (2) and (3).  The second term is not greater or equal, it's less

6 195 In Eq. (4), You should use the natural logarithm, not the decimal logarithm.

3 104 "The instrument used in this paper" -> "The instrument used in this study"

4 144 "Minimum stopband attenuation (dB) -44" in table 2 and "Stopband minimum attenuation/dB 44" in table 3 are seems to be the same item, but the terminology and the sign are different.

4 145 If Eq. (1) is not the author's formula, then a reference is required.

4 146 "Frequency" -> "frequency"

5 185 "in the time domain and frequency domain" -> "in the time and frequency domain"

6 211 "this paper uses the sym3 wavelet" what is "sym3"?

8 275 "Sigmoid" -> "sigmoid"

10 307 "R7-5800HCPU" -> "R7-5800H CPU"

10 319 is dataset2 after preprocessing?

11 338 What is 'loss value'?

Round 2

Reviewer 2 Report

 The manuscript entitled “Debris flow infrasound recognition method based on improved LeNet5 network”, by X. Leng, L. Feng, O. Ou, X. Du, D. Liu, & X. Tang, presents an improved and good work.

The manuscript should be acceptable for publication in the present form.

Author Response

We thank you very much for your positive comment on our manuscript.